

# The sensitivity of snowfall to weather states over Sweden

Lars Norin[1], Abhay Devasthale[1], and Tristan S. L'Ecuyer[2]

[1]Atmospheric Remote Sensing Unit, Research Department, Swedish Meteorological and Hydrological Institute, Norrköping, Sweden
[2]Department of Atmospheric and Oceanic Sciences, University of Wisconsin-Madison, Madison, Wisconsin, USA

*Correspondence to:* Lars Norin (lars.norin@smhi.se)

**Abstract.** For a high latitude country like Sweden snowfall is an important contributor to the regional water cycle. Furthermore, snowfall impacts surface properties, affects atmospheric thermodynamics, has implications for traffic and logistics management, disaster preparedness, and also impacts climate through changes in surface albedo and turbulent heat fluxes. For Sweden it has been shown that large-scale atmospheric circulation patterns, or weather states, are important for precipitation variability.

Although the link between atmospheric circulation patterns and precipitation has been investigated for rainfall there are no studied focused on the sensitivity of snowfall to weather states over Sweden.

    In this work we investigate the response of snowfall to eight selected weather states. These weather states consist of four dominant wind directions together with cyclonic and anti-cyclonic circulation patterns and enhanced positive and negative phases of the North Atlantic oscillation. The presented analysis is based on multiple data sources, such as ground-based

radar measurements, satellite observations, spatially-interpolated in situ observations, and reanalysis data. The data from these sources converge to underline the sensitivity of falling snow over Sweden to the different weather states.

    In this paper we examine both average snowfall intensities and snowfall accumulations associated with the different weather states. It is shown that even though the heaviest snowfall intensities occur during conditions with winds from the southwest, the largest contribution to snowfall accumulation arrives from winds from the southeast. Large differences in snowfall due

to variations in the North Atlantic oscillation are shown as well as a strong effect of cyclonic and anti-cyclonic circulation patterns. Satellite observations are used to reveal the vertical structures of snowfall during the different weather states.

## 1 Introduction

At high latitudes, solid precipitation is an important contributor to the regional water cycle (Levizzani et al., 2011; Waliser et al., 2011), it impacts surface properties, affects atmospheric thermodynamics, has implications for traffic and logistics man-

agement, disaster preparedness, and also impacts climate through changes in surface albedo and turbulent heat fluxes. In a country like Sweden with strong north-south gradients in meteorological, topographical, and surface conditions, understanding the response of snowfall to large-scale weather states is even more important. Such a characterisation also provides guidance concerning the performance of numerical weather prediction models by quantifying their strengths and limitations in wintertime regimes.



Precipitation variability over Sweden has been investigated in selected previous studies (Busuioc et al., 2001; Johansson and Chen, 2003; Hellström, 2005; Gustafsson et al., 2010; Devasthale and Norin, 2014). Using data from 33 ground-based stations between the years 1980–1990, Busuioc et al. (2001) found that the link between precipitation variability and the dominant circulation patterns can be very complex over Sweden. For example, while the North Atlantic oscillation (NAO) is the principle

mode of sea-level pressure variability during cold months and correlates strongly (0.86) with precipitation, it explained only 32 % to 54 % of precipitation variability locally. Hellström (2005) pointed out the importance of cyclonic and anti-cyclonic circulations in regulating the summertime extreme precipitation events while Gustafsson et al. (2010) further argued for the crucial role of continental Europe and the Baltic Sea in controlling the moisture transport during extreme precipitation events. But the majority of these studies focus on liquid precipitation (rain) and the summer months. Far fewer studies directly address

the factors that govern snowfall and its covariability with the other regional impacts of the different wintertime weather states that occur across Sweden and those that do lack sufficient detail to inform forecasts.

Snowfall over Sweden can occur under the influence of a range of circulations patterns (here termed "weather states"). Using Lamb classification (Lamb, 1950), Chen (2000) investigated the monthly circulation climatology over Sweden. He showed that among 25 different circulation patterns observed over Sweden, four states dominate. These can be characterised by either

cyclonic or anti-cyclonic conditions or persistent large-scale westerly or southwesterly wind regimes. During winter months, westerly and southwesterly winds occur 21 % and 17 % respectively, followed by anti-cyclonic conditions that occur about 11 % of the time. Using a similar classification methodology, Linderson (2001) further observed a robust correspondence between local precipitation and large-scale circulation during winter months over southern Sweden. She showed that over southern Sweden anti-cyclonic conditions dominated (16 %) all circulation types. More recently, Thomas and Devasthale

(2014) used this weather state classification to investigate temperature and humidity response and pollutant variability over Scandinavia. They showed that enhanced positive phase of the NAO and southeasterly winds warm the entire troposphere and the surface. These studies form the basis for the investigation and selection of weather states used in the present study.

The main aim of this work is to exploit the synergy between two remote sensing systems, ground-based weather radar and space-based cloud radar to further document the character of falling snow during each weather state and to quantify the

influence of the key meteorological factors that modulate snowfall differences between the different weather states. We seek to answer two specific questions:

1. How does falling snow respond to different atmospheric weather states? The aim here is to investigate to what extent snowfall is sensitive to weather states (different wind directions, mean sea level pressures (MSLP), and phases of the NAO) and to characterise differences in this dependence between southern and northern parts of the country. The latter is

30       especially important since the snowfall distribution and frequency is very inhomogeneous meridionally across Sweden (Busuioc et al., 2001; Devasthale and Norin, 2014).

2. What are the typical meteorological conditions associated with each weather state and how do they contribute to the observed differences in snowfall between weather states? Here we quantify the extent to which similar co-variability between meteorological parameters and snowfall exists across all weather states.



## 2 Datasets and weather states

In this work we have investigated the response of snowfall to eight different weather states. These weather states consist of four dominant wind direction (northeast, northwest, southeast, and southwest) as well as conditions of high and low MSLP and enhanced positive or negative phases of the NAO. To study the response of snowfall to the different weather states we made

use of a combination of spatially-interpolated in situ observations, reanalyses, ground-based radar measurements, and satellite observations. All dates satisfying any of these conditions were identified in reanalyses. Mean radar-derived surface snowfall intensities and meteorological analyses of surface air temperatures were then composited for these dates. Finally, vertical profiles of reflectivity from satellite observations were extracted to document the vertical structure and implied microphysical properties of snowfall in each weather state.

Below follows a description of all datasets used in this study together with a definition of the selected weather states.

### 2.1 ERA-Interim reanalysis

ERA-Interim provides global atmospheric reanalysis, updated continuously since 1979 (Dee et al., 2011). Gridded data products include a large variety of 3-hourly surface parameters, describing weather as well as ocean-wave and land-surface conditions, and 6-hourly upper-air parameters covering the troposphere and stratosphere. Vertical columns of monthly averages for

many atmospheric parameters and other derived fields are also available. Berrisford et al. (2009) provide a detailed description of the ERA-Interim archive.

The data assimilation system used to produce ERA-Interim is based on a 2006 release of the European Centre for Medium-range Weather Forecasting (ECMWF) Integrated Forecast Model. The system includes a 4-dimensional variational analysis with a 12-hour analysis window. The spatial resolution of the data set is approximately 80 km on 60 vertical levels from the

surface up to 0.1 hPa.

In this study u and v components of winds at 850 hPa were used to identify the dominant weather states. The wind fields have a spatial resolution of about 80 km and are available every 3 hours.

### 2.2 Mesan

To characterise the phase of precipitation 2 m air temperatures were obtained from Mesan (Häggmark et al., 1997, 2000), a

mesoscale analysis system that has been operational at the Swedish Meteorological and Hydrological Institute (SMHI) since 1996. Mesan creates spatially continuous fields of selected meteorological parameters (such as precipitation, temperature, humidity, visibility, wind, and clouds) on a model grid using observations from synoptic stations, automatic stations, satellites, and weather radars. In addition to these observations, a first guess field from SMHI's operational numerical weather prediction model, HIRLAM (Källén, 1996), is used. The field values of the meteorological parameters are calculated in Mesan using the

optimal interpolation technique (Daley, 1993). The Mesan 2 m temperatures are further adjusted for both the fraction of land in the first guess field's grid square and for the altitude of the temperature observations.



Mesan covers Scandinavia and the Baltic Sea and has a spatial resolution of $0.1° \times 0.1°$ ($11\,\text{km} \times 11\,\text{km}$). The temporal resolution is one hour. Archived data exist from 1998 to present and is continuously updated.

## 2.3 Nordrad

The meteorological offices in the Nordic countries (Danish Meteorological Institute, Estonian Environment Agency, Finnish

Meteorological Institute, Latvian Environment, Geology and Meteorology Agency, Norwegian Meteorological Institute, and SMHI) collaborate by exchanging radar data in real time. The collaboration is called Nordrad (Carlsson, 1995) and consists today of 35 operational weather radars.

    The SMHI uses data from all available weather radars within Nordrad to generate composite radar images covering the Nordic countries. These data are corrected for beam blockage (Bech et al., 2003). Non-meteorological echoes, originating

for example from ground clutter or clear-air targets, are removed by a satellite-based filter (Michelson, 2006). Furthermore, in order to correct the radar measurements for distance-dependence, the radar data is adjusted to precipitation measurements from rain gauges by fitting a second degree polynomial to the logarithmic gauge-to-radar ratio (Michelson and Koistinen, 2000). The conversion from reflectivity to precipitation rate $R$ ($\text{mm}\,\text{h}^{-1}$) is made using the empirical relationship $Z = aR^b$ (e.g. Battan, 1973), where $a = 200$ and $b = 1.5$. For a more detailed description of the radar processing see, e.g., Michelson (2006); Norin

et al. (2015).

    The quality-adjusted radar data are merged into a precipitation composite, Nordrad, which is generated every $15\,\text{min}$ and has a spatial resolution of $2\,\text{km} \times 2\,\text{km}$. Archived Nordrad composite images exist at the SMHI from 2005 to present. However, in late 2007 a hardware upgrade was installed at the Swedish weather radars to enable Doppler processing for all scans. In September 2014 a modernisation process began to upgrade all Swedish weather radars to dual polarisation. As a result, the

present analysis is restricted to the homogeneous processing period from January 2008 through December 2012.

## 2.4 Snowfall product from the CloudSat Cloud Profiling Radar

CloudSat snowfall estimates were obtained from the Release 04 2C-SNOW-PROFILE (hereafter, 2CSNOW) data product (Wood, 2011; Wood et al., 2013). The product provides vertically-resolved snowfall estimates consistent with profiles of W-band (94 GHz) reflectivities measured by CloudSat's Cloud Profiling Radar (CPR). CloudSat observes falling snow between

$82°$ N and $82°$ S latitude along a ground track that repeats every $16\,\text{days}$ providing moderately dense spatial sampling at high latitudes relative to more equatorial locations (Kulie et al., 2016; McIlhattan and L'Ecuyer, 2016). Snowfall rates are produced at the full $1.7\,\text{km}$ along-track by $1.4\,\text{km}$ cross-track spatial resolution of the CPR. The radar has an intrinsic vertical resolution of $485\,\text{m}$, but measurements are oversampled to yield profiles at an effective vertical resolution of $239\,\text{m}$.

    Rather than assuming a fixed relationship between reflectivity and snowfall rate (a so-called $Z$–$S$ relationship), the 2CSNOW

retrieval algorithm estimates vertical profiles of the probability density functions of snow particle size distribution parameters. Information collected during recent field campaigns are used to supply a priori estimates of the environmental distributions of these parameters as well as snow particle microphysical and scattering properties in the retrieval (Wood et al., 2015). Attenuation and multiple-scattering by snow particles are also modelled. A snowfall rate is generated whenever the CloudSat





2C-PRECIP-COLUMN product (Haynes et al., 2009) indicates the presence of frozen precipitation at the surface defined as a scene with a melted mass fraction of less than $10\,\%$. The melted mass fraction is estimated based on the height of the freezing level and a melting model that accounts for the environmental lapse rate below the freezing level derived from ECMWF analyses.

Ground clutter affects the CPR measurements in the radar range bins nearest the surface so these near-surface bins are not explicitly included in the reflectivity profiles when retrievals are performed, creating what is sometimes called a blind zone. Consequently 2CSNOW estimates the surface snowfall rate as the rate retrieved in the radar bin immediately above the blind zone. Over land, this zone extends about $1\,\mathrm{km}$ above the surface (Smalley et al., 2014). This surface snowfall rate is assigned a confidence value from "None" to "High" depending on the expected performance of the forward model and the reliability of
the temperature-based estimate of the precipitation phase, among other factors.

### 2.5   Humidity data from the Atmospheric Infrared Sounder

Humidity profiles from the Atmospheric Infrared Sounder (AIRS) instrument flying onboard NASA's Aqua satellite since 2002 are further used to investigate differences in the moisture advection during weather states (Chahine et al., 2006; Susskind et al., 2014; Devasthale et al., 2016). Aqua leads the A-Train constellation of satellites wherein CloudSat follows closely behind
the Aqua satellite in the same orbit. The retrievals provided in the standard daily gridded Version 6 product (AIRX3STD) at $850\,\mathrm{hPa}$ are analysed (Susskind et al., 2014). The estimated accuracy of humidity profiles is 15 % per $2\,\mathrm{km}$. In this study, AIRS provides an independent information on the moisture variability during the different weather states.

### 2.6   Weather states

In the present study we have, following Thomas and Devasthale (2014), selected eight different weather states which frequently
occur over the Nordic countries. The eight weather states are divided into two groups: wind regimes, consisting of conditions dominated by four wind directions (northwest, northeast, southeast and southwest), and synoptic states, during which either cyclonic or anti-cyclonic conditions or enhanced positive or negative phases of the NAO prevailed.

To find days corresponding to the first four weather states the average daily wind speed and wind direction at $850\,\mathrm{hPa}$ from the ERA-Interim reanalysis were extracted. Based on these daily averages, days during which a particular wind direction
prevailed and persisted for at least $3\,\mathrm{days}$ were selected in order to capture large-scale precipitation events. Cyclonic and anti-cyclonic conditions were found in a similar way by analysing the MSLP over central Sweden. Anti-cyclonic conditions were identified when the MSLP was higher than one standard deviation for 3 consecutive days. If the MSLP was lower than one standard deviation, cyclonic conditions prevailed. The NAO indices were obtained from NOAA's Climate Prediction Center website (NOAA, 2016). Enhanced positive and negative phases of NAO correspond to absolute values of the NAO
index exceeding one standard deviation during those phases. Enhanced positive or negative NAO conditions were defined as 3 consecutive days with an enhanced positive or negative NAO index.





Examples of mean circulation patterns for the weather states characterised by dominant wind directions are shown in Fig. 1 whereas circulation patterns for high and low MSLP together with enhanced positive and negative phases of the NAO are displayed in Fig. 2.

## 3   Data processing

The data sets from ERA-Interim and Mesan extend from 1979 and 1998 to present, respectively. Nordrad data are nominally available since 2005 but due to modifications to the Swedish weather radar network, a homogeneous Nordrad dataset can only be compiled from 2008 to 2014. Data from CloudSat are available from 2006 to present. Thus in this study we have restricted the study period to 1 January 2008 to 31 December 2012 when all required datasets are available.

As described in Norin et al. (2015) a simple but robust radar-based snowfall product can be generated by combining Nordrad's precipitation composite with 2 m temperatures from Mesan. For every Nordrad composite image the 2 m temperature field from Mesan nearest in time and space was used to identify frozen precipitation. If the 2 m temperature was below freezing (0°C) the corresponding precipitation from Nordrad was classified as snow, otherwise as rain. In this work we have used snowfall estimates from the Nordrad composites but in order to provide a uniformly calibrated data set as well as to have full knowledge of the measurements we have only selected radar data originating from areas covered by Swedish radars.

Measurements from ground-based radars generally have lower quality far away from the radar station. Comparing ground-based radar measurements of snowfall intensities to CloudSat snowfall measurements, Norin et al. (2015) found that Nordrad snowfall estimates are optimal at distances between 46–82 km from a Swedish radar station. Snowfall intensities from Nordrad were therefore selected within this optimal distance range from the 12 Swedish weather radar stations together with the corresponding 2 m temperatures from Mesan, for each of the 8 identified weather states. The average number of radar cells within the optimal measurement zone for any radar is approximately 3620 while Mesan has approximately 120 data gridboxes covering the same area.

Average snowfall intensity and the average 2 m temperature were calculated for all days assigned to each weather state. Corresponding temperature anomalies were defined relative to the daily average 2 m temperatures from Mesan for all regions within the optimum distance over the period from 2008–2012.

Each day CloudSat passes over Sweden twice and samples data every 1.4 km. The CloudSat satellite has been operational since 2006 but in April 2011, the satellite experienced a battery anomaly that caused a ten month gap in observations. Measurements resumed in February 2012 but the satellite has only operated during the daytime portion of its orbit since that time. In this study, all available ascending and descending passes of CloudSat from 1 January 2006 to 31 December 2012 in the latitude band between 54° N and 70° N were used (excluding the summer months, June to September, in which no snowfall was observed). To ensure high quality, only CloudSat snowfall retrievals with a confidence flag corresponding to "Moderate" or "High" were used in the analysis yielding between 0 and 1007 observed snowfall profiles per day. CloudSat vertical snowfall profiles together with the estimated surface height over Sweden were composited for each weather state to characterise



the vertical structure of snowfall in each regime. Snowfall intensities below 1250 m above the surface were excluded to avoid CloudSat's blind zone.

The humidity profiles at 850 hPa available at the $1° \times 1°$ resolution from the AIRX3STD were analysed in the same way as the surface temperature data from MESAN, i.e. by taking into account the only those cases where surface temperatures (also derived from AIRS for the sake of consistency) were colder than $0°$ C. The data from the ascending passes of AIRS were analysed.

# 4 Results and discussion

The response of snowfall to the eight selected weather states were examined using the snowfall products from Nordrad and CloudSat, together with temperature observations from Mesan and measurements of specific humidity from AIRS. Below, the results for the two different groups of weather states are presented and the implications for snowfall accumulations are discussed.

## 4.1 Snowfall during dominant wind regimes

Average snowfall intensity for the weather states characterised by the four dominant wind directions (northwest, northeast, southwest, and southeast, as illustrated in Fig. 1) from all of Sweden's 12 weather radars are shown in Fig. 3a. In general the average snowfall intensities are seen to decrease with increasing latitude. The smallest average snowfall intensities, below $0.2 \, \mathrm{mm \, h^{-1}}$, are seen to occur for winds from the southeast at northern radar stations (above $60°$ N). It is also found that the largest average snowfall intensities occur when winds originate from the southwest. This is valid for all radar stations, except for one radar located in the northwestern region which is slightly more influenced by northwesterly frontal systems arriving from the Norwegian Sea. For the southernmost radar stations (below $59°$ N), where this effect is most evident, average snowfall intensities are near $0.5 \, \mathrm{mm \, h^{-1}}$.

The corresponding 2 m temperatures, shown in Fig. 3c, reveal that the most intense snowfall occurs at warmer temperatures, ranging from approximately $-1°$ C to $-4°$ C, whereas the lightest snowfall intensities correspond to colder temperatures, between $-6°$ C and $-11°$ C. For northerly winds the snowfall intensities are similar, around $0.15$–$0.25 \, \mathrm{mm \, h^{-1}}$. However, snowfall intensities are slightly larger for winds from the northwest for most radar stations when corresponding 2 m temperatures are on average $1$–$3°$ C warmer compared to the northeasterly wind regime. Figure 3e shows that temperature anomalies during snowfall are, in general, largest for the southernmost stations (below $59°$ N), for all wind directions. However, the largest temperature anomalies for all station occur for winds from the northeast, ranging from $-2°$ C to $-7°$ C while the smallest temperature anomalies ($0°$ C to $-3°$ C) occur for winds from the southeast.

Average values of specific humidity, observed by AIRS, are shown in Fig. 4. In agreement with the snowfall observations it is seen that winds from the southwest advect sufficient moisture over all of Sweden which in combination with warm temperatures lead to large snowfall intensities. During winds from the southeast, moist air originating from continental Europe passes over the southern and central parts of Sweden. Together with warm temperatures this leads to increased snowfall intensities. On the





other hand, this moist air does not reach the northern parts of the country and this lack of moisture in combination with cold temperatures results in small snowfall intensities. For northerly winds moist air from the Atlantic is blocked by the mountains on the border to Norway and the result is moderate snowfall over Sweden. Anomalies of specific humidity are presented in Fig. 5. The anomalies show that unusually moist air covers Sweden during southwesterly winds while the opposite occurs

during winds from the north. During winds from the southeast, southern Sweden receives moist air while the air in the northern part of the country is drier than usual.

Figure 6 shows the vertical structure of snowfall, as observed by CloudSat, for the four weather states characterised by dominant wind directions. Mean snowfall rates as a function of altitude are shown together with 2D histograms of snowfall intensities and altitudes. In addition, the altitudes at which 50 % and 95 % of the snowfall rates were observed are displayed.

It is seen that the largest mean near-surface snowfall rates, above $0.2 \, \mathrm{mm \, h^{-1}}$, occur during westerly wind regimes. These snow events are characterised by deeper snow columns and stronger vertical intensity gradients that may be indicative of enhanced aggregation processes relative to easterly snow events. The smallest mean near-surface snowfall intensity, below $0.1 \, \mathrm{mm \, h^{-1}}$, occurs during winds from the southeast, consistent with the observations from the ground-based radars. It is also interesting to note that the mean snowfall intensity (approximately $0.1 \, \mathrm{mm \, h^{-1}}$) from the ground up to $5 \, \mathrm{km}$ above the surface

in northeasterly regimes indicating an absence of aggregation in the prevailing colder, drier conditions. Near the surface, the distribution of snowfall intensities peaks around $0.5 \, \mathrm{mm \, h^{-1}}$ when winds are from the southwest. For winds from the east, the distribution of snowfall rates peaks at a much lower intensity, below $0.01 \, \mathrm{mm \, h^{-1}}$.

The different response of the snowfall intensities to these weather states can be understood by examining the circulation patterns in Fig. 1. Winds from the southwest originate from the Atlantic and the North Sea bringing moist, warm air over

Sweden resulting in heavy snowfall once the temperature falls and sustains below $0°\,\mathrm{C}$. Winds from the northwest also originate from the Atlantic but pass over the mountainous border to Norway before arriving in Sweden, thereby bringing less snow at slightly colder temperatures compared to southwesterly winds. These northwesterly winds have the second largest influence on the central Swedish regions (after southwesterly winds). Winds from the northeast, on the other hand, arrive from the Norwegian/Barent's Sea and bring relatively colder air and less moisture, have the second largest impact (after southwesterly

winds) on the northernmost Swedish regions. Finally, southeasterly winds originating from continental Europe pick up moisture from the southern Baltic Sea resulting in moderate snowfall intensities over southern Sweden. Their influence on the northern parts of the country is minimal. This is mainly due to the fact that southeasterly frontal systems dry up before reaching the northernmost parts of Sweden. Furthermore, the Bay of Bothnia is frozen and covered by sea ice during core winter months closing the moisture source in the Baltic Sea. These results underline the varied meridional nature of the response of snowfall

to dominant wind directions.

## 4.2  Snowfall during different synoptic states

The principles behind the results for the dominant wind regimes are also manifested in the observed sensitivity of snowfall to MSLP and enhanced positive or negative NAO index.



Figure 3b shows that for the weather state characterised by anti-cyclonic conditions (high MSLP), corresponding to an atmospheric blocking-like circulation pattern that stagnates over the entire southern Scandinavia, average snowfall intensities are similar for all radar stations, around $0.2$–$0.25\,\mathrm{mm\,h^{-1}}$. Temperatures range between $-6^\circ$ C to $-2^\circ$ C with the colder temperatures occurring at the northern stations (Fig. 3d, f). When cyclonic conditions prevail (low MSLP) the snowfall intensities are higher, as expected, ranging from $0.25$ to $0.40\,\mathrm{mm\,h^{-1}}$ for the southern radar stations. On the other hand, at the northern stations snowfall rates are $0.2$–$0.3\,\mathrm{mm\,h^{-1}}$ for both cyclonic and anti-cyclonic regimes. The average values for the specific humidity, as observed by AIRS, for high and low MSLP are presented in Fig. 7a and b. The corresponding anomalies are shown in Fig. 8. It is seen that during high MSLP the air is more humid than for conditions with low MSLP. This is in contrast to the observations of snowfall intensities. To understand this apparent paradox we must examine the vertical structure of snowfall.

Figure 9a and b show the vertical structure of snowfall during high and low MSLP, respectively, as observed by CloudSat. It can be seen that the mean snowfall intensity near the surface is around $0.2\,\mathrm{mm\,h^{-1}}$ during both weather states, agreeing with ground-based observations (cf. Fig. 3b). CloudSat observations suggest, however, that more vertically-developed snowfall events occur predominantly during anti-cyclonic conditions. When MSLP is high, snowfall rates are confined around $0.1$–$0.2\,\mathrm{mm\,h^{-1}}$ at all heights from the surface up to $3\,\mathrm{km}$ whereas the vertical distributions of snowfall during low MSLP events are more widespread, ranging from $0.01\,\mathrm{mm\,h^{-1}}$ to $0.5\,\mathrm{mm\,h^{-1}}$. This explains why the specific humidity is higher during anti-cyclonic conditions while at the same time the surface snowfall rates are slightly lower.

The apparent insensitivity of snowfall to change in MSLP over the northern parts of Sweden can be understood by noting that during anti-cyclonic conditions these areas receive similar amount of average snowfall from winds carrying moisture from the northern part of the Norwegian Sea. Very low differences in average temperatures over the northern regions during anti-cyclonic and cyclonic conditions are further observed (Fig. 3d). At $850\,\mathrm{hPa}$ the air is more humid for high MSLP leading to the observed differences in the vertical structure of snowfall. However, even though the distribution of snowfall intensities are very different during these weather states the average snowfall rates are similar at the surface.

Ground-based observations of snowfall intensities during weather states with enhanced positive NAO indices, shown in Fig. 3b, range between $0.20$–$0.35\,\mathrm{mm\,h^{-1}}$ and are quite similar to those observed during enhanced negative NAO phase. Except for the southeastern coast, snowfall intensities during positive NAO indices are slightly larger. The corresponding average temperatures are shown in Fig. 3d. It is seen that temperatures decrease with increasing latitude but also that temperatures are $1$–$3^\circ$ C colder for enhanced negative phase of the NAO. The average values for the specific humidity are depicted in Fig. 7c and d. During enhanced positive NAO conditions the air is more humid compared to conditions of enhanced negative NAO.

By examining CloudSat's observations of the vertical structure of snowfall during these weather states, shown in Fig. 9c and d, it can be seen that the mean snowfall rate near the surface is slightly higher for the enhanced positive NAO phase, near $0.25\,\mathrm{mm\,h^{-1}}$, compared to the enhanced negative NAO phase, $0.2\,\mathrm{mm\,h^{-1}}$. The median heights of the snowfall intensities are similar for both conditions. A more obvious difference can be seen in the distribution of the snowfall intensities. For enhanced positive NAO phase the snowfall distribution peaks around $0.4\,\mathrm{mm\,h^{-1}}$ for heights from the surface up to $2\,\mathrm{km}$ whereas the distribution of snowfall intensities during enhanced negative NAO phase is more widespread, dominated by snowfall intensities from $0.005\,\mathrm{mm\,h^{-1}}$ to $0.1\,\mathrm{mm\,h^{-1}}$.



The similarity in snowfall intensities during enhanced negative and enhanced positive NAO phases can be explained by temperature anomalies (Fig. 3f) and circulation patterns (Fig. 2). During the enhanced positive NAO phase, only the persistent southwesterly winds can bring sufficient moisture to colder northern parts of Sweden to create snowfall events with average intensities similar to those during the negative phase of NAO. While in the case of enhanced negative NAO phase over the
southern parts of country, the circulation pattern favours conditions wherein southerly moist winds are confronted by colder and drier northerly winds (Fig. 8d) resulting in snowfall events with average intensities similar to those during the enhanced positive NAO phase. These results underline the complex interplay between circulation patterns and snowfall by affecting moisture transport and meridional temperature gradients.

### 4.3   Implications for snowfall accumulations

The relative strengths of the snowfall intensities for the different weather states are shown in Fig. 10a and b. In Fig. 10a it can be seen that the highest snowfall intensities occur for winds from the southwest for almost all stations in Sweden, which is a result of the warm temperatures and moist air that characterises this weather state. The other dominant wind directions all lead to similar snowfall intensities, albeit due to different reasons which were discussed in the preceding sections.

Even though snowfall intensities are highest for winds from the southwest this weather state does not contribute most to the
total snowfall amount. The relative contribution to snowfall accumulation for weather states characterised by dominant wind directions are shown in Fig. 10c. Here it is seen that for southern radar stations easterly winds lead to the largest amount of snow whereas for northern stations winds from the northwest lead to the largest snowfall accumulation. These results reflect the frequency with which snowfall occur during the different wind regimes. For example, even though southwesterly winds are the most common wind direction in Sweden they are usually accompanied by warm temperatures. Hence precipitation normally
falls as rain during this weather state but if temperatures happen to be below $0°$ C, the snowfall intensities become large. On the other hand, during easterly winds temperatures are more often freezing and these weather states therefore contribute most to the total accumulation of snow.

For the other weather states (high and low MSLP and enhanced positive or negative NAO indices) the relative strength of the snowfall intensities are shown in Fig. 10b. The absolute differences in snowfall intensities are smaller for these weather states
compared to those characterised by the dominant wind directions. However, it is seen that higher snowfall intensities occur for low MSLP for all Sweden (except at the very north) while snowfall intensities are slightly higher for enhanced positive NAO for almost the entire country, except for the southernmost part.

For high and low MSLP it is clear that the contribution to total snowfall accumulation is larger for weather states with low MSLP for all stations. This is partly a reflection of the higher snowfall intensities associated with low MSLP but also due to
the higher frequency with which it snows during these conditions. For enhanced positive and negative NAO indices it is clear that in southern Sweden the contribution to total snow accumulation is dominated by conditions with enhanced negative NAO. This is because snowfall is relatively uncommon during weather states characterised by enhanced positive NAO, due to the warm temperatures. For northern Sweden both weather states contribute with approximately the same amount.



## 5  Conclusions

In this work we have examined the sensitivity of snowfall to eight different atmospheric circulation patterns, or weather states, commonly occurring over Sweden. The circulation patterns include four dominant wind directions, high and low MSLP, and enhanced positive and negative NAO indices. To this end we have used a combination of different data sources: spatially-interpolated in situ observations, ground-based radar data, satellite measurements, and reanalyses products.

It was found that falling snow over Sweden is indeed sensitive to different atmospheric weather states. A clear variation in average snowfall intensity was observed, depending on circulation pattern but also depending on latitude. Snowfall over southern Sweden responds to different weather states in a different way compared to over the northern parts of the country. The largest snowfall intensities were observed for southwesterly winds, which bring warm, moist air from the Atlantic over the entire country. The smallest average snowfall rates for northern Sweden were observed during southeasterly winds while the southern parts of the country received the smallest snowfall rates northeasterly winds.

Using satellite observations the vertical snowfall structure of snowfall were examined. Clear differences between the various circulation patterns could be observed. During easterly winds snowfall was seen to be much more shallow compared to snowfall during westerly winds. During high MSLP and positive phases of the NAO snowfall tended to be concentrated to larger intensities compared to conditions with low MSLP or negative phases of the NAO.

Snowfall accumulations depend on the observed snowfall rates but also on the frequency with which the weather states occur. It was found that easterly winds contribute most to snowfall accumulation over southern Sweden whereas westerly winds bring the largest snow amounts to the northern parts of the country. Further, low MSLP and negative phases of the NAO contribute more to snowfall accumulation than high MSLP and positive phases of the NAO.

These results indicate a complex interplay between circulation patterns, heat and moisture transport, and snowfall and underlines the varied meridional nature of this interplay. The study also demonstrates the advantage of the synergetic use of different observing systems to investigate snowfall variability.

## 6  Data availability

ERA-Interim data can be accessed from the website of the ECMWF, `apps.ecmwf.int/datasets/`. Nordrad data for research are available on request from SMHI. Composites based only on Swedish radars, very similar to Nordrad, are available from `opendata-catalog.smhi.se/explore/` where also Mesan data can be downloaded. CloudSat data are available from `cloudsat.atmos.colostate.edu/data`.

*Author contributions.* The authors designed the study and prepared the manuscript together. L. Norin and A. Devasthale performed the data analyses.

*Competing interests.* The authors declare that they have no conflict of interest.





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





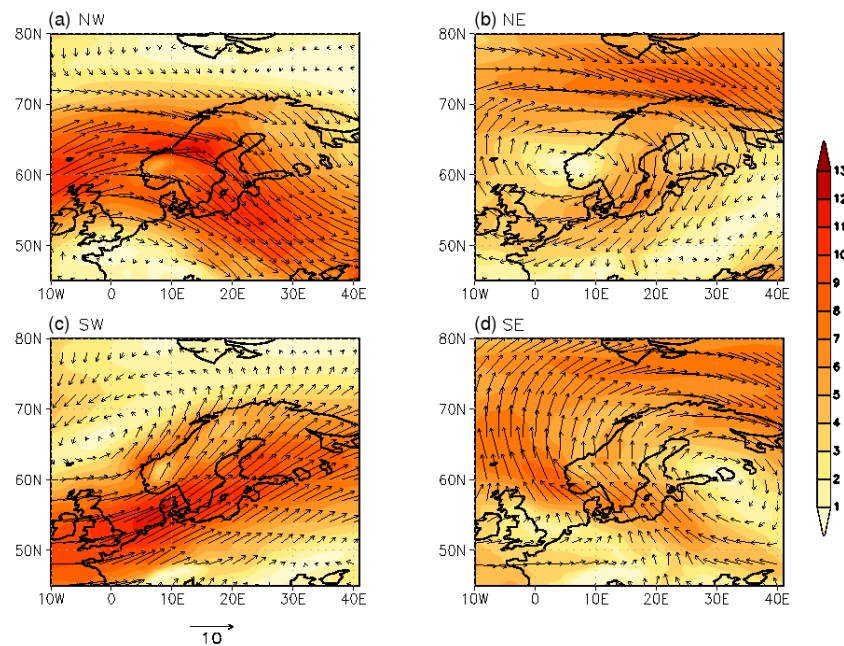

**Figure 1.** Atmospheric circulation patterns at 850 hPa leading to four persistent wind directions over Sweden. Panels **(a)** to **(d)** show circulation patterns resulting in winds from northwest, northeast, southwest, and southeast, respectively. The colourbar indicates wind strength (in m s$^{-1}$).





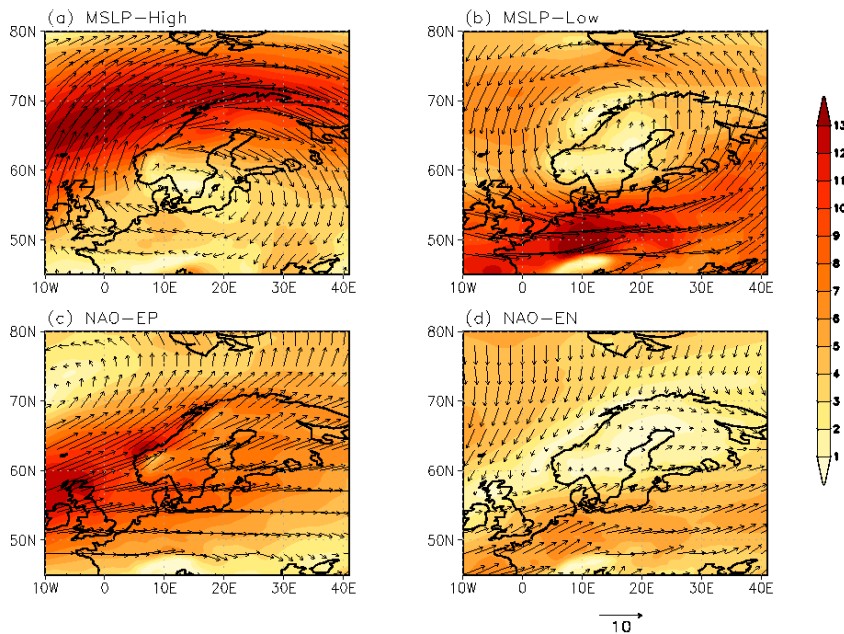

**Figure 2.** Atmospheric circulation patterns at 850 hPa for four different weather states. Panels **(a)** and **(b)** show circulation patterns during high and low MSLP conditions whereas panels **(c)** and **(d)** depict patterns for enhanced positive and negative phases of the NAO. The colourbar shows wind strength (in $\mathrm{m\,s^{-1}}$).





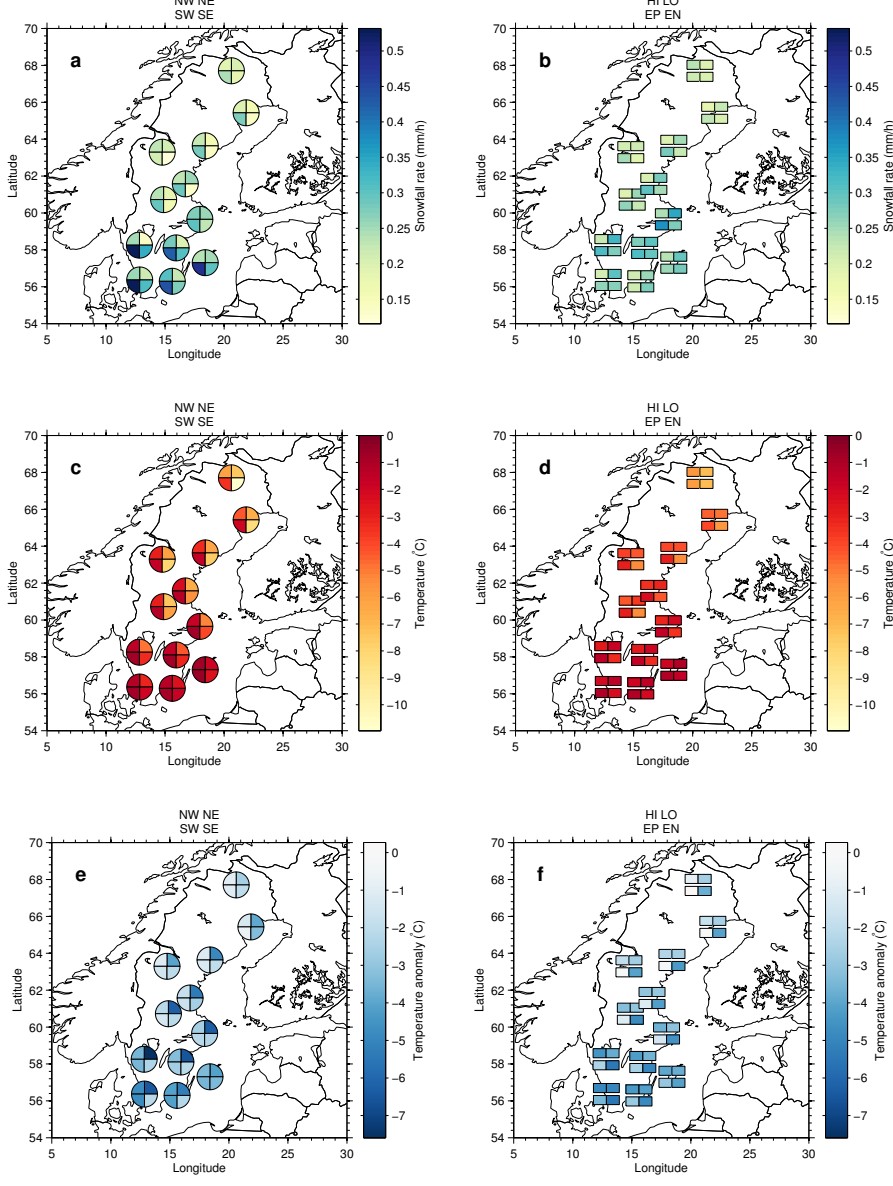

**Figure 3.** Average snowfall intensities (in $\mathrm{mm\,h^{-1}}$) observed by the Swedish weather radar network for winds from northwest, northeast, southwest, and southeast **(a)** and during high and low MSLP and enhanced positive and negative NAO conditions **(b)**. Average 2 m temperatures (in °C), obtained from the mesoscale analysis system Mesan, for winds from northwest, northeast, southwest, and southeast **(c)** and during high and low MSLP and enhanced positive and negative NAO conditions **(d)**. Average temperature anomalies (in °C), obtained from the mesoscale analysis system Mesan, for winds from northwest, northeast, southwest, and southeast **(e)** and during high and low MSLP and enhanced positive and negative NAO conditions **(f)**.





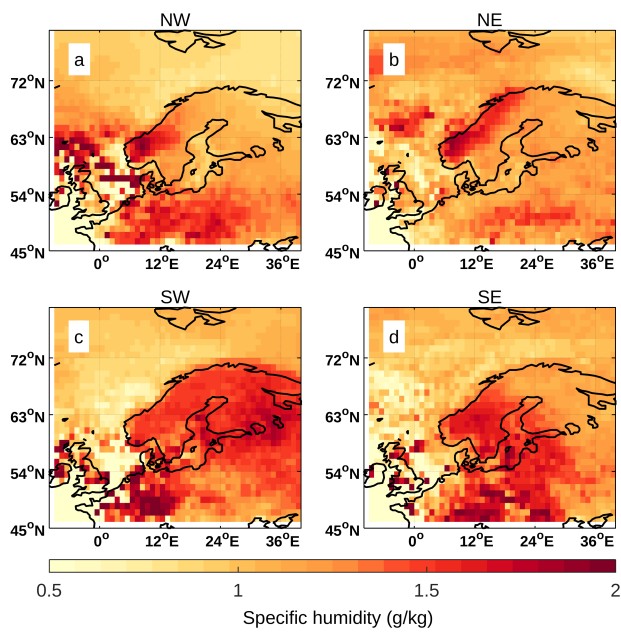

**Figure 4.** Average values of specific humidity, observed by AIRS at 850 hPa. Panel **(a)** shows the specific humidity for atmospheric conditions dominated by winds from the northwest. In panels **(b)** to **(d)** the winds are from northeast, southwest and southeast, respectively.



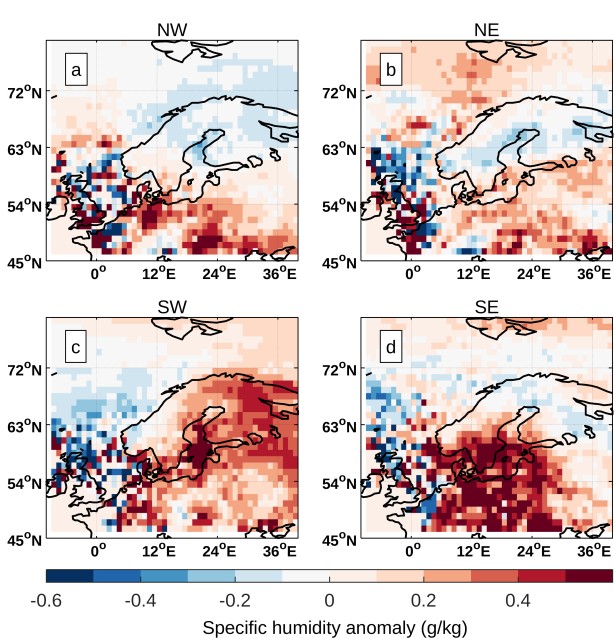

**Figure 5.** Anomalies of specific humidity, observed by AIRS at 850 hPa. Panel **(a)** shows anomalies in the specific humidity for atmospheric conditions dominated by winds from the northwest. In panels **(b)** to **(d)** the winds are from northeast, southwest and southeast, respectively.





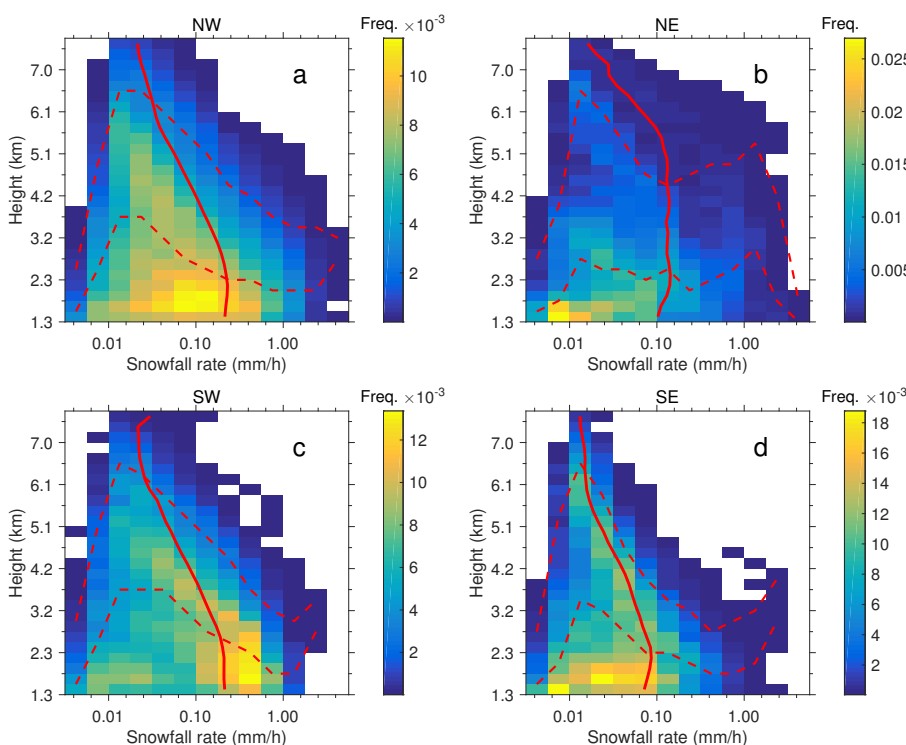

**Figure 6.** Histograms showing relative frequency of snowfall rates as a function of altitude above the surface, using data from CloudSat. The different panels show the vertical structure of snowfall for weather states dominated by different wind directions (northwest, northeast, southwest, and southeast). The red solid lines show the mean snowfall rates as a function of altitude. The two red dashed lines show the altitudes where 50 % and 95 % of the snowfall have occurred, as a function of snowfall intensity.





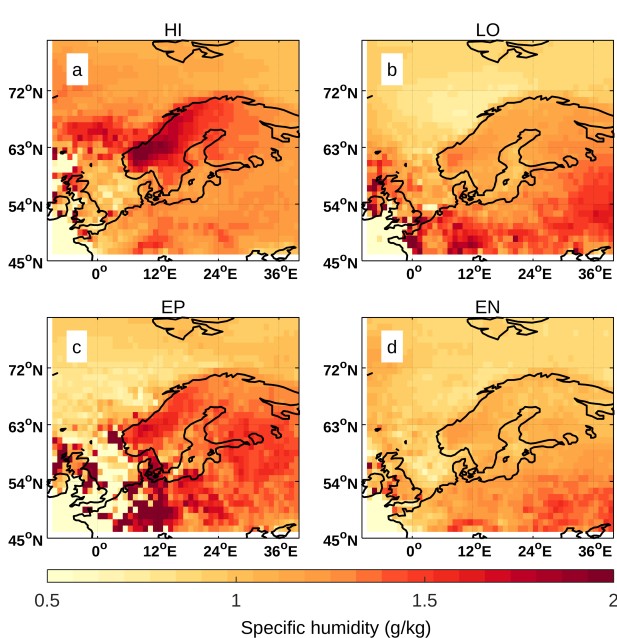

**Figure 7.** Average values of specific humidity, observed by AIRS at 850 hPa. Panels **(a)** and **(b)** show average values of the specific humidity for high and low MSLP while panels **(c)** and **(d)** show the corresponding variable for enhanced positive and negative phases of the NAO.





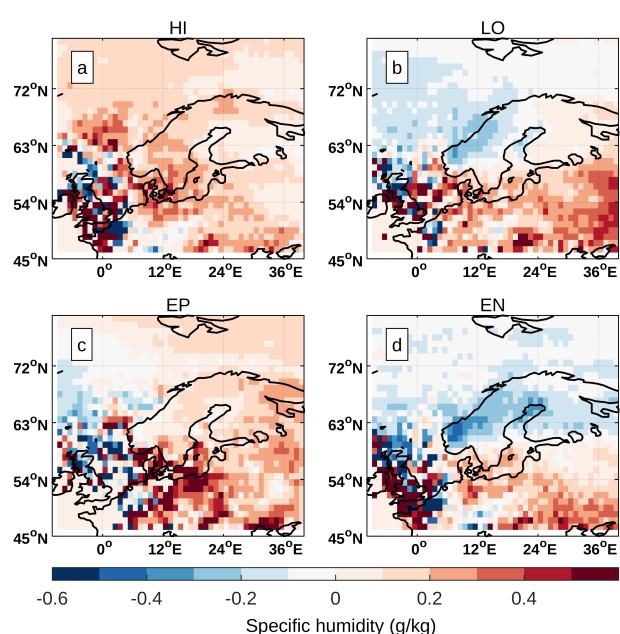

**Figure 8.** Anomalies of specific humidity, observed by AIRS at 850 hPa. Panels **(a)** and **(b)** show anomalies in the specific humidity for high and low MSLP while panels **(c)** and **(d)** show the corresponding variable for enhanced positive and negative phases of the NAO.





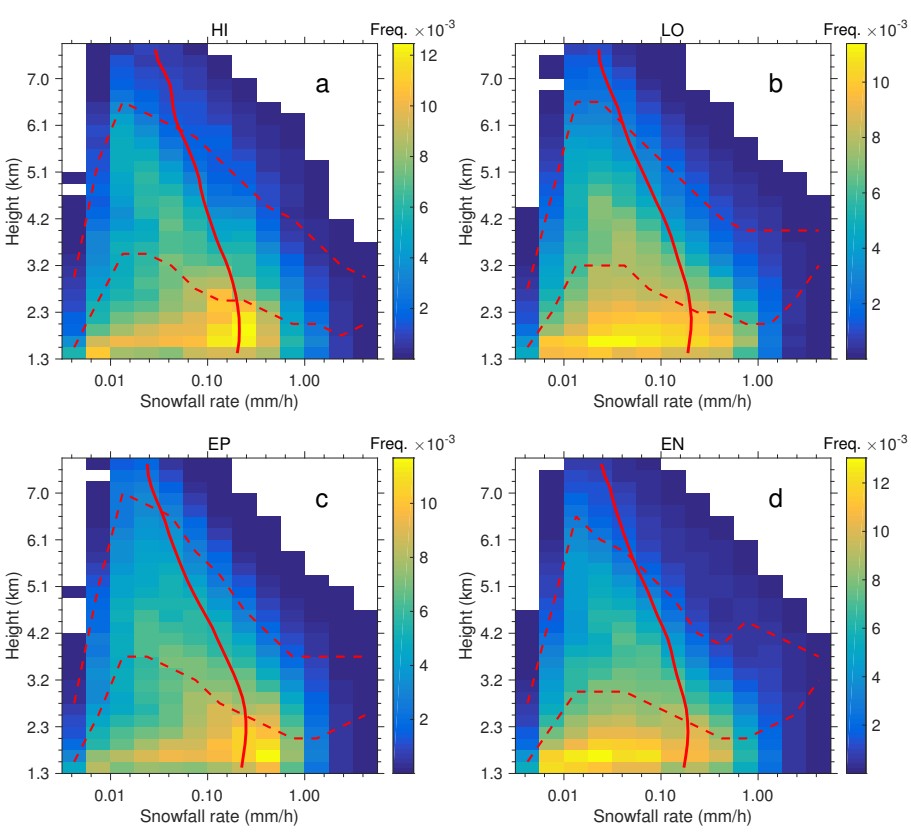

**Figure 9.** Histograms showing relative frequency of snowfall rates as a function of altitude above the surface, using data from CloudSat. The different panels show the vertical structure of snowfall for weather states with high and low MSLP (**a** and **b**), and enhanced positive and negative NAO indices (**c** and **d**). The red solid line shows the mean snowfall rate as a function of altitude. The two red dashed lines show the altitudes where 50 % and 95 % of the snowfall have occurred, as a function of snowfall intensity.



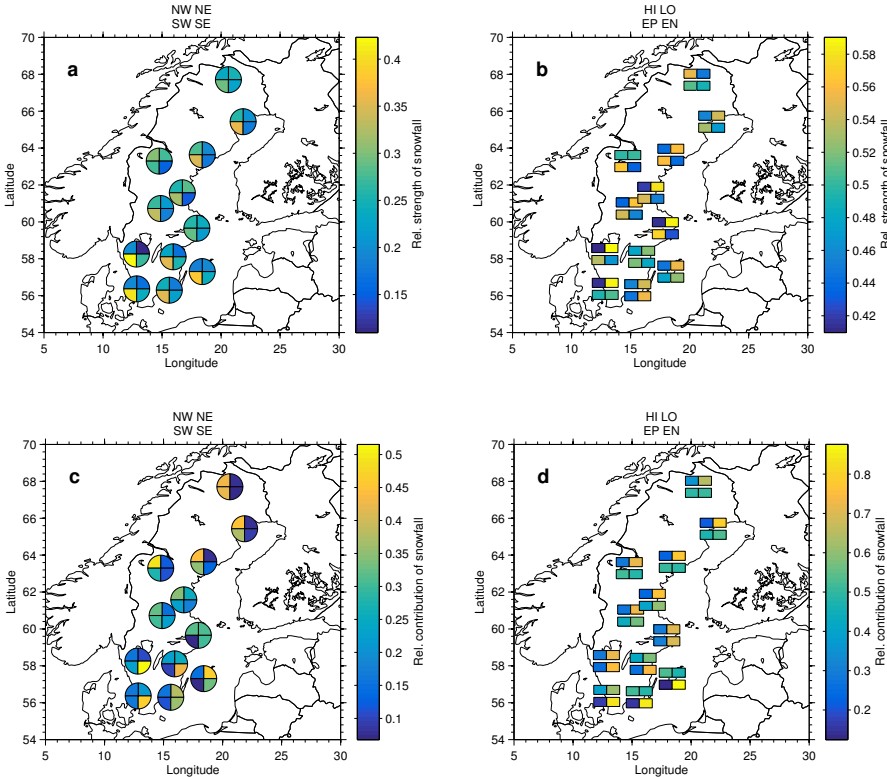

**Figure 10.** Relative strength of snowfall intensities for winds from northwest, northeast, southwest, and southeast **(a)**. For every radar station the sum of the relative strength from the four wind directions is equal to one. Relative strength of snowfall intensities during high and low MSLP and enhanced positive and negative NAO conditions **(b)**. Here the sum of the relative strength during high and low MSLP for every station is equal to one as well as the sum of the relative strength during enhanced positive and negative NAO conditions is equal to one. Relative contribution of total snowfall for winds from northwest, northeast, southwest, and southeast **(c)**. The sum for every radar station is equal to one. Relative contribution of total snowfall during high and low MSLP and enhanced positive and negative NAO conditions **(d)**. The sum of the relative contribution during high and low MSLP for every station is equal to one and the sum of the relative contribution during enhanced positive and negative NAO conditions is equal to one.