# Peer review of "The sensitivity of snowfall to weather states over Sweden"

_Atmospheric Measurement Techniques, 2017_

## Referee Comment (RC1) · Anonymous Referee #1 · 11 Mar 2017

This paper describes an analysis of the dependence of snowfall in Sweden on large-scale patterns of atmospheric circulation. The effects of prevailing wind direction, high or low pressure as well as positive and negative North Atlantic Oscillation (NAO) are considered.

The analysis methods the authors use appear sound, and the for the most part, the conclusions are consistent with the presented data. The study only concerns Sweden and as such has little applicability for other regions, but it could have an impact on understanding the origin of snowfall events in that particular region, and affect societal readiness for responding to those events. The manuscript is generally clearly written; there are minor grammatical issues here and there, but these have minimal impact on

the readability and can be fixed in copyediting, so I will not comment further on them here.

I think that the remarks given below should be addressed before the study is published.

General comments to the authors:

1. This seems like a precipitation science paper, but has been submitted to a journal that generally deals with advances to methods of atmospheric measurement. If you can make a few additions that steer the paper slightly in that direction - for example by describing how your methodology could (or could not) be applied in analysis of other snowy regions - that would make it a better fit for the journal.

2. In a few parts of the paper, the effects of the frequency/intensity of snowfall in a given weather state on one hand, and the frequency of that weather state on the other, are mixed. Maybe you could add a table that shows the relative frequencies of the different weather states; that would help decouple those effects.

3. Related to point 2, the statistical analysis is somewhat muddled by the fact that the occurrences of the different weather states are likely not independent. The prevailing winds, the NAO index and the occurrence of high/low pressure systems are probably correlated with each other at least to some degree. This should also be quantified, and if there are strong correlations, the impact of those on the results should be discussed.

4. The analysis relies on the "prevailing" wind and pressure conditions, but Sweden is a big country and there can be quite some variability in those conditions between the southern and northern extremes. For example, Figs. 1b and 1d show that there can be about 90 deg difference in the 850 hPa wind direction between the north and the south. In fact, in "northeasterly" wind conditions in Fig. 1b, the prevailing winds in the far north appear to be from the northwest! And if one follows the streamlines in that figure, it appears that the flow to most of Sweden appears to originate from the Norwegian Sea in the (north)west rather than from the Barents Sea in the northeast.

[Figure]

I wonder how much effect this has on the results, especially as the authors discuss differences between northern and southern Sweden in several places in the paper.

5. Since there are already studies that investigated the connections of rainfall patterns to weather states, it would be reasonable for the authors to devote a paragraph or two to discussing the similarities and differences that can be found between the responses of rain and snow events to large-scale weather states.

Specific comments:

Page 2:

Line 1: "Selected" previous studies? Should this say "several" instead?

Page 5:

Lines 7-8: Did you investigate if there were systematic differences in surface snowfall between 2CSNOW and the ground-based radar?

Line 14: I think OCO-2 is currently the A-Train lead satellite.

Lines 27-28: This should probably say "more than one standard deviation *above/below the mean*"

Page 6:

Line 22: I think that the average is only for cases where it is snowing, i.e. zero snowfall is not included in the average? Or am I mistaken? In any case, this should be clarified.

Page 7:

Lines 23-25: "However, snowfall intensities are..." This is a confusing sentence, I am not sure what you are trying to say.

Line 25: Temperature anomalies relative to what?

Page 8:

Line 3: See my previous comment: what is the reference for calculating specific humidity anomalies?

Page 9:

Line 16: Comparing the mean lines Figs. 9a and 9b, it is rather hard to see any difference between them near the surface.

Page 10:

Line 10: "Relative strengths of snowfall" - again, relative to what?

Figs. 1 and 2: Since the wind speed is already encoded in the arrows, the using the color to also show that is somewhat redundant. On the other hand, the pressure patterns might be interesting. How about using color to show the pressure instead?

---

## Referee Comment (RC2) · Anonymous Referee #2 · 19 May 2017

Review of "The sensitivity of snowfall to weather states over Sweden"

The goals of the paper are well stated, the analysis is appropriate for achieving the goals, and the writing and figures are clear. I do have some questions about how the analysis was performed.

General comments

1. Previous research has identified the 8 states the paper analyzes. While investigating snowfall relative to these states is appropriate, I think the authors should note that these states are not independent from one another. I would like to know how many events comprise each category, and if individual dates can be included in multiple categories.

2. Are there no snowfall observations over Sweden that can be used to further validate the remotely sensed data? At times the authors refer to "snowfall accumulation". As I understand the analysis performed, no statements can be made about snowfall accumulation, as the duration of the snowfall events does not appear to be considered.

3. Figures 1 and 2 – are these composites (means) of all events? Or individual events that are representative of the composites?

4. On the use of CloudSat – As you discuss, data represents a "snapshot" taken once or twice per day. Is snowfall occurring during all of these snapshots? How many snapshots are there per event? I am concerned about how representative these snapshots are for the events analyzed using the other more frequent data sources, as comparisons are made between them.

5. I would like to know more about how the wind directions were defined. The NAO is straightforward and I can see how identifying the presence of a pressure center over land is tractable. But wind direction can be highly variable over the domains shown in the figures.

6. I think it would be useful to the reader to see figures that show the radar coverages referred to in the text (48-82 km from a Swedish radar station), so we can have an idea of how representative the data is with respect to the entire country.

7. Related to the above comment, were the intensities for each event defined by averages over each radar volume? How many times were used in each event?

8. I see that Norin et al. (2015) describes how the snowfall product is generated. Given the fact that your conclusions are heavily dependent upon the quality of the data that results from the methods used to define the product, I think it would be worthwhile to further discuss the limitations of the radar-based data.

9. Section 4.2, line 15. I see what you are referring to in Figure 9a and b, but I am not sure how this resolves the apparent paradox discussed in the previous paragraph. Perhaps you could be more explicit.

Minor comments

1. In line 5, it is stated that the NAO correlates strongly with precipitation, but explains only 32 to 54% of precipitation variability locally (Busuioc et al. 2001). In line 17, Linderson (2001) further observes a robust correspondence between local precipitation and large-scale circulation during winter months over southern Sweden. As written, it seems these results are contradictory. If this is the case, it should be highlighted as motivation for this study. If this isn't the case, perhaps the text could more clearly explain the differences in the studies.

2. Line 22 – Such a characterization also provides guidance concerning the performance of NWP models by quantifying their strengths and limitations in wintertime regimes – could you state more clearly how your work relates to NWP verification?

3. Line 30 – snowfall distribution and frequency is very inhomogeneous meridionally across Sweden. How does it vary?

---

## Author Comment (AC1) · 28 Jun 2017

We thank the referee for the time and effort devoted to review this manuscript as well as for the very constructive comments and suggestions. Below, please find a point-by-point reply to the comments (reproduced in italics).

*General comments to the authors:*

*1.   This seems like a precipitation science paper, but has been submitted to a journal that generally deals with advances to methods of atmospheric measurement. If you can make a few additions that steer the paper slightly in that direction — for example by describing how your methodology could (or could not) be applied in*

[Figure]

*analysis of other snowy regions — that would make it a better fit for the journal.*

We agree that the manuscript is more scientific in nature. But one of the main reasons for submitting in AMTD was to demonstrate the usefulness of long-term radar observations in Sweden and how the synergy with the latest space-based observations (specifically CloudSat) can be exploited to study snowfall. We believe such study can also be done for the other regions that have long-term radar observations.

*2. In a few parts of the paper, the effects of the frequency/intensity of snowfall in a given weather state on one hand, and the frequency of that weather state on the other, are mixed. Maybe you could add a table that shows the relative frequencies of the different weather states; that would help decouple those effects.*

We thanks the referee for the suggestion. A table listing the number of days containing the different weather states (as well as how many other other weather states occurred for those days) has been added to the revised manuscript.

*3. Related to point 2, the statistical analysis is somewhat muddled by the fact that the occurrences of the different weather states are likely not independent. The prevailing winds, the NAO index and the occurrence of high/low pressure systems are probably correlated with each other at least to some degree. This should also be quantified, and if there are strong correlations, the impact of those on the results should be discussed.*

As mentioned above, a table listing dates with simultaneous weather states has been added. We agree that there are correlations between some of the different weather states. However, within the different groups of weather states (wind directions, high and low MSLP, and different NAO indices) there are no overlapping dates. Nevertheless, the stronger correlations are pointed out in the revised manuscript.

*4. The analysis relies on the "prevailing" wind and pressure conditions, but Sweden is a big country and there can be quite some variability in those conditions between the southern and northern extremes. For example, Figs. 1b and 1d show that there can be about 90 deg difference in the 850 hPa wind direction between the north and the south. In fact, in "northeasterly" wind conditions in Fig. 1b, the prevailing winds in the far north appear to be from the northwest! And if one follows the streamlines in that figure, it appears that the flow to most of Sweden appears to originate from the Norwegian Sea in the (north)west rather than from the Barents Sea in the northeast. I wonder how much effect this has on the results, especially as the authors discuss differences between northern and southern Sweden in several places in the paper.*

The weather states were calculated in a paper by Thomas and Devasthale (Atmos. Chem. Phys., 14, 11545–11555, 2014). In their paper they evaluate the weather states using measurements from an area positioned over southern Sweden. This is the reason why certain wind directions can appear differently in the northernmost part of the country. The description of their methodology has been expanded in the manuscript. The area in which the weather states were calculated has been added to figures 1 and 2.

*5. Since there are already studies that investigated the connections of rainfall patterns to weather states, it would be reasonable for the authors to devote a paragraph or two to discussing the similarities and differences that can be found between the responses of rain and snow events to large-scale weather states.*

Following the referee suggestion, a paragraph discussing difference in circulation regimes during summer and winter seasons is added in the revised manuscript. Perhaps the most comprehensive evaluation of monthly circulation types over Sweden is done by Chen (2000) and Busuioc et al. (2001). Chen (2000) has shown that while the large-scale circulation patterns over North East Atlantic favouring westerly and northwesterly winds dominate 35–50% of the time during the summer months (June

through August), the wintertime circulation patterns are characterized by nearly equal dominance of cyclonic and anticyclonic conditions. Similarly, Busuioc et al. (2001) showed that during winter months (especially beginning of year), NAO explained 41% of MSLP variability followed by the dipole structure with centres located over eastern Scandinavia and East Atlantic explained about 24% variability in January and the third mode represented cyclonic/anticyclonic conditions. However, during the summer rainfall months, the cyclonic/anticyclonic conditions dominated circulation types representing about 37% variability.

*Specific comments:*

*Page 2:*
*Line 1: "Selected" previous studies? Should this say "several" instead?*

Done.

*Page 5:*
*Lines 7-8: Did you investigate if there were systematic differences in surface snowfall between 2CSNOW and the ground-based radar?*

Yes, a comparison of the surface snowfall from 2CSNOW and Nordrad was investigated and described in the paper by Norin et al. (Atmos. Meas. Tech., 8, 5009–5021, 2015). A short paragraph has been added to further describe the results of that study.

*Line 14: I think OCO-2 is currently the A-Train lead satellite.*

True. This is now corrected in the manuscript.

*Lines 27-28: This should probably say "more than one standard deviation \*above/below*

*the mean\*"*

Done.

*Page 6:*
*Line 22: I think that the average is only for cases where it is snowing, i.e. zero snowfall is not included in the average? Or am I mistaken? In any case, this should be clarified.*

Correct. This is now clarified.

*Page 7:*
*Lines 23-25: "However, snowfall intensities are..." This is a confusing sentence, I am not sure what you are trying to say.*

The sentence has been rewritten.

*Line 25: Temperature anomalies relative to what?*

The temperatures anomalies are relative to the average temperature for the investigated years. This is now clarified in the manuscript.

*Page 8:*
*Line 3: See my previous comment: what is the reference for calculating specific humidity anomalies?*

The specific humidity anomalies are relative to the average specific humidity for the investigated years. This is now clarified in the manuscript.

*Page 9:*
*Line 16: Comparing the mean lines Figs. 9a and 9b, it is rather hard to see any differ-*

*ence between them near the surface.*

Agreed. The sentence has been rewritten.

*Page 10:*
*Line 10: "Relative strengths of snowfall" — again, relative to what?*

Here relative snowfall means that the sum of all four snowfall rates are equal to one. A sentence has been added to clarify this.

*Figs. 1 and 2: Since the wind speed is already encoded in the arrows, the using the color to also show that is somewhat redundant. On the other hand, the pressure patterns might be interesting. How about using color to show the pressure instead?*

We thank the referee for the suggestion. Although it would likely be an improvement it is unfortunately not possible for us within this project.

---

## Author Comment (AC2) · 28 Jun 2017

We thank the referee for the time and effort spent to review the manuscript as well as for the very constructive comments and suggestions. Below, please find a point-by-point reply to the comments (reproduced in italics).

**General comments**

1. Previous research has identified the 8 states the paper analyzes. While investigating snowfall relative to these states is appropriate, I think the authors should note that these states are not independent from one another. I would like to know how many events comprise each category, and if individual dates can be included in

**multiple categories.**

A table listing the number of days containing the different weather states (as well as how many other other weather states occurred for those days) has been added to the manuscript.

2. Are there no snowfall observations over Sweden that can be used to further validate the remotely sensed data? At times the authors refer to "snowfall accumulation". As I understand the analysis performed, no statements can be made about snowfall accumulation, as the duration of the snowfall events does not appear to be considered.

We agree that no quantitative statements can be made about snowfall accumulation. To find a qualitative estimation of the relative contribution of the different weather states to the total snowfall amount, the average snowfall intensities were multiplied with the number of days for each weather state. The text in Section 4.3 has been adjusted in the revised manuscript.

3. Figures 1 and 2 — are these composites (means) of all events? Or individual events that are representative of the composites?

These are individual events, representative of the composites. This has now been clarified in the manuscript.

4. On the use of CloudSat — As you discuss, data represents a "snapshot" taken once or twice per day. Is snowfall occurring during all of these snapshots? How many snapshots are there per event? I am concerned about how representative these snapshots are for the events analyzed using the other more frequent data sources, as comparisons are made between them.

Yes, only passes during which CloudSat detected snowfall over Sweden were

used in the analysis. The number of data points with identified snowfall per pass varied from 1 up to 1007. It is true that measurements from the ground-based radars are more frequent as well as made over spatially smaller areas. However, only CloudSat can provide a detailed vertical structure of the snowfall. The impact of CloudSat's lower temporal and spatial resolution is now discussed in the manuscript.

5. I would like to know more about how the wind directions were defined. The NAO is straightforward and I can see how identifying the presence of a pressure center over land is tractable. But wind direction can be highly variable over the domains shown in the figures.

The method for identifying the weather states was presented in a paper by Thomas and Devasthale (Atmos. Chem. Phys., 14, 11545—11555, 2014). The wind directions were determined by analysing the average wind direction over an area near the centre of Sweden. For this reason wind direction can be different when far from this area (e.g. in the northernmost part of the country). The description of this method has now been expanded in the manuscript. The area in which the weather states were calculated has been added to figures 1 and 2.

6. I think it would be useful to the reader to see figures that show the radar coverages referred to in the text (48–82 km from a Swedish radar station), so we can have an idea of how representative the data is with respect to the entire country.

We thank the referee for the suggestion. The radar coverages used in the analysis have now been added to figure 3 and 10.

7. Related to the above comment, were the intensities for each event defined by averages over each radar volume? How many times were used in each event?

СЗ

Yes, for each radar composite the average snowfall intensity was determined using all radar cells detecting snowfall. All snowfall measurements detected during each day (24 h) with an identified weather state were used.

8. I see that Norin et al. (2015) describes how the snowfall product is generated. Given the fact that your conclusions are heavily dependent upon the quality of the data that results from the methods used to define the product, I think it would be worthwhile to further discuss the limitations of the radar-based data.

A paragraph discussing the limitations of the radar-based snowfall product has been added to the revised manuscript.

9. Section 4.2, line 15. I see what you are referring to in Figure 9a and b, but I am not sure how this resolves the apparent paradox discussed in the previous paragraph. Perhaps you could be more explicit.

Fig. 2a shows that during High MSLP conditions, anticyclonic winds coming from Northern Europe further pick up moisture from the North Sea before making precipitation over western coast of Norway and southern parts of Sweden. This leads to increase in specific humidity over these regions (Figs. 7a and 8a). These snowfall events, whenever they do occur, also seem to be well vertically developed as evident in Fig. 9a. However, during low MSLP conditions, the location of low pressure (Fig. 2b) is such that more northerly, colder winds result in lower specific humidity anomalies (Fig. 8b). Notice however that over the southernmost tip of Sweden, humidity anomalies are slightly positive during low MSLP conditions due to warmer southwesterly winds. Since the centre of low pressure is slightly northward (compared to the centre of high pressure), the occurrence of snowfall events with differing intensities, over northern and southern regions of Sweden leads to more broader vertical distribution of snowfall as observed by CloudSat (Fig. 9b).

**Minor comments**

1. In line 5, it is stated that the NAO correlates strongly with precipitation, but explains only 32 to 54% of precipitation variability locally (Busuioc et al. 2001). In line 17, Linderson (2001) further observes a robust correspondence between local precipitation and large-scale circulation during winter months over southern Sweden. As written, it seems these results are contradictory. If this is the case, it should be highlighted as motivation for this study. If this isn't the case, perhaps the text could more clearly explain the differences in the studies.

We agree that the current formulation of these sentences is confusing. They are therefore clarified in the revised manuscript. During winter months, NAO can explain a significant part of precipitation variability. But in a year as a whole, the other circulation types, such as cyclonic systems, persistent westerly winds and even anticyclonic systems, can be more important, especially over the southern parts of Sweden.

2. Line 22 — Such a characterization also provides guidance concerning the performance of NWP models by quantifying their strengths and limitations in wintertime regimes — could you state more clearly how your work relates to NWP verification?

This rather abstract sentence is also rephrased in the revised manuscript. Our intention was to point out the fact that NWP models need to show similar response of snow to weather states as we have presented in this study, in order to capture snowfall events realistically. Therefore, the observationally based results from our study can be used for process oriented evaluation of NWP models to test the fidelity of NWP models.

3. Line 30 — snowfall distribution and frequency is very inhomogeneous meridionally across Sweden. How does it vary?

The snowfall distribution across Sweden is shown in Fig. 1 below. The southern parts of Sweden receives the smallest amounts of snow. The snow depth increases steadily with increasing latitudes but the largest snow depths are found in the mountains to the west, along the border to Norway. This description has been added to the manuscript.

Fig. 1. Average largest snow depth during the period 1961–1990.